# Anticoagulation Status and Left Atrial Appendage Occlusion Indications in Hospitalized Cardiology Patients with Atrial Fibrillation: A Hellenic Cardiorenal Morbidity Snapshot (HECMOS) Sub-Study

**DOI:** 10.3390/medicina59101881

**Published:** 2023-10-23

**Authors:** Dimitris Tsiachris, Panteleimon E. Papakonstantinou, Ioannis Doundoulakis, Panagiotis Tsioufis, Michail Botis, Kyriakos Dimitriadis, Ioannis Leontsinis, Athanasios Kordalis, Christos-Konstantinos Antoniou, Emmanouil Mantzouranis, Panagiotis Iliakis, Panayotis K. Vlachakis, Konstantinos A. Gatzoulis, Konstantinos Tsioufis

**Affiliations:** 1First Department of Cardiology, National and Kapodistrian University, “Hippokration” Hospital, 11527 Athens, Greece; pantelispapakon@gmail.com (P.E.P.); doudougiannis@gmail.com (I.D.); ptsioufis@gmail.com (P.T.); mgmpotis94@gmail.com (M.B.); dimitriadiskyr@yahoo.gr (K.D.); giannisleontsinis@gmail.com (I.L.); akordalis@gmail.com (A.K.); ckantoniou@hotmail.gr (C.-K.A.); mantzoup@gmail.com (E.M.); panayiotisiliakis@gmail.com (P.I.); vlachakispanag@gmail.com (P.K.V.); kgatzoul@med.uoa.gr (K.A.G.); ktsioufis@gmail.com (K.T.); 2Athens Heart Center, Athens Medical Center, 15125 Athens, Greece

**Keywords:** atrial fibrillation, heart failure, rhythm strategy, catheter ablation, snapshot

## Abstract

*Background and Objectives*: The proper use of oral anticoagulants is crucial in the management of non-valvular atrial fibrillation (AF) patients. Left atrial appendage closure (LAAC) may be considered for stroke prevention in patients with AF and contraindications for long-term anticoagulant treatment. We aimed to assess anticoagulation status and LAAC indications in patients with AF from the HECMOS (Hellenic Cardiorenal Morbidity Snapshot) survey. *Materials and Methods*: The HECMOS was a nationwide snapshot survey of cardiorenal morbidity in hospitalized cardiology patients. HECMOS used an electronic platform to collect demographic and clinically relevant information from all patients hospitalized on 3 March 2022 in 55 different cardiology departments. In this substudy, we included patients with known AF without mechanical prosthetic valves or moderate-to-severe mitral valve stenosis. Patients with prior stroke, previous major bleeding, poor adherence to anticoagulants, and end-stage renal disease were considered candidates for LAAC. *Results*: Two hundred fifty-six patients (mean age 76.6 ± 11.7, 148 males) were included in our analysis. Most of them (*n* = 159; 62%) suffered from persistent AF. The mean CHA2DS2-VASc score was 4.28 ± 1.7, while the mean HAS-BLED score was 1.47 ± 0.9. Three out of three patients with a a CHA2DS2-VASc score of 0 or 1 (female) were inappropriately anticoagulated. Sixteen out of eighteen patients with a CHA2DS2-VASc score 1 or 2 (if female) received anticoagulants. Thirty-one out of two hundred thirty-five patients with a CHA2DS2-VASc score > 1 or 2 (if female) were inappropriately not anticoagulated. Relative indications for LAAC were present in 68 patients with NVAF (63 had only one risk factor and 5 had two concurrent risk factors). In detail, 36 had a prior stroke, 17 patients had a history of major bleeding, 15 patients reported poor or no adherence to the anticoagulant therapy and 5 had an eGFR value < 15 mL/min/1.73 m^2^ for a total of 73 risk factors. Moreover, 33 had a HAS-BLED score ≥ 3. No LAAC treatment was recorded. *Conclusions*: Anticoagulation status was nearly optimal in a high-thromboembolic-risk population of cardiology patients who were mainly treated using NOACs. One out of four AF patients should be screened for LAAC.

## 1. Introduction

Atrial fibrillation (AF) is the most frequently encountered cardiac arrhythmia, with an increasing trend worldwide [1]. The association between AF and cardioembolic stroke is well established. Additionally, cardioembolic strokes are associated with increased mortality and a greater socioeconomic burden compared with non-cardioembolic strokes [2]. Anticoagulation therapy is a cornerstone treatment for stroke prevention in atrial fibrillation (AF) [3,4]. The proper use of oral anticoagulants (OACs) is crucial in the management of AF [5]. Available therapeutic regimens include vitamin K antagonists and, more recently, non-vitamin K antagonists (NOACs). NOACs were a revolution in anticoagulation therapy for AF, providing more predictable effects with a rapid onset and offset of their action and fewer drug and food interactions while requiring less frequent laboratory monitoring [6]. Even in the era of NOACs, the compliance of AF patients with OAC therapy reported in some studies is still poor, and discontinuation rates persist as an issue [7]. The anticoagulation agent prescribed and the patient’s ethnicity are reported to affect compliance [8]. A patient’s frailty status, high bleeding risk, severe renal impairment, and unwillingness to receive anticoagulation therapy are some of the main reported issues in patients who did not receive an OAC from their treating physicians [3,4,5,9].

The left atrial appendage is considered the most common location for thrombus formation in patients with non-valvular AF (AF patients without mechanical prosthetic valves and/or moderate-to-severe mitral stenosis). Recently, percutaneous left atrial appendage closure has emerged as a safe therapeutic alternative in patients with increased stroke risk and ineligibility for the administration of an OAC [10,11]. According to the 2020 ESC guidelines for the management of AF, left atrial appendage closure may be considered for stroke prevention in patients with AF and contraindications for long-term anticoagulant treatment [3]. However, beyond the current recommendations [3,4], in a contemporary, real-world dataset of hospitalized patients with comorbid, non-valvular AF, almost one out of six hospitalized patients may be considered eligible for left atrial appendage closure due to either a history of major bleeding or a history of embolic stroke under oral anticoagulation [5,12]. In this study, we sought to analyze a contemporary cohort of hospitalized patients with comorbid AF. Our principal objective was to assess anticoagulation status and delineate left atrial appendage closure indications in patients with “nonvalvular” AF from the HECMOS (Hellenic Cardiorenal Morbidity Snapshot) survey.

## 2. Materials and Methods

This study is an ancillary analysis from the HECMOS multicenter cross-sectional observational snapshot survey. Detailed descriptions of the study population, design and results have been reported. Briefly, the HECMOS study sought to investigate the contemporary trends in cardiorenal morbidity among hospitalized patients in cardiology wards across Greece on an ordinary weekday, 3 March 2022 [13]. The First Cardiology Clinic of the National and Kapodistrian University of Athens (NKUA) organized the study in collaboration with the Second and Third NKUA Cardiology Clinics under the auspices of INAKEN (the Institute for Study, Research, and Education on Vascular, Heart, Brain, and Kidney Disease).

Patients were recruited from all the cardiological departments of the Hellenic National Public Health System, as well as from high-volume centers in the private sector. In total, patients from 55 different departments nationwide were enrolled, providing a representative population sample οf the county’s geographical distribution. The study population consisted of inpatients aged >18 years who were able to provide informed consent independently or via their legal representative.

Detailed information regarding each patient’s admission type was obtained. Admission was categorized as scheduled or emergency admission. In the case of emergency admission, comprehensive information regarding the patients’ precipitating symptoms were collected, including dyspnea, dizziness, a loss of consciousness, thoracic pain, edema, and palpitations. The cardiovascular causes of admission were stratified to include heart failure exacerbation, tachyarrhythmias, bradyarrhythmias, acute coronary syndrome, pulmonary embolism, syncope, inherited cardiomyopathies, channelopathies, pericarditis, pulmonary hypertension and severe valvular heart disease, with or without the presence of comorbid heart failure.

Data collection included the patients’ demographics, anthropometric measurements including weight and height, and self-reported habits, including alcohol consumption and smoking status. Details regarding the participants’ comorbidities, including the presence of chronic heart failure, AF, hypertension, bleeding history, chronic kidney disease, severe liver dysfunction, chronic obstructive pulmonary disease, obstructive sleep apnea syndrome and diabetes mellitus, were obtained. Patients suffering from diabetes mellitus were further classified as having type I diabetes, type II diabetes or prediabetes. The presence and type of cardiac implanted electronic devices were recorded. Specifically, the dataset included the presence and type of cardiac pacemakers (single-chamber or dual-chamber), the presence of implantable cardioverter-defibrillators and the presence of cardiac resynchronization therapy with or without a concurrent defibrillator (CRT-D and CRT-P, respectively). Electrocardiographic, echocardiographic, and laboratory parameters, concurrent to the snapshot survey, were acquired. The main echocardiographic features were the left ventricle ejection fraction (LVEF), left ventricular end-diastolic diameter, right ventricular performance and measures of pulmonary artery pressures, as well as the presence of a mechanical or bioprosthetic heart valve. Laboratory features of interest included hemoglobulin, the lipidemic profile and iron studies, as well markers of myocardial injury, natriuretic peptides and electrolyte status (sodium and potassium). The eGFR was calculated according to the 2021 CKD-EPI equation. Participants were also asked about their influenza and COVID-19 vaccination status.

The pharmacological regimens administered to the patients were also recorded. Specifically, the dataset included information about beta blockers, renin–angotensin–aldosterone system inhibitors, angiotensin receptor–neprilysin inhibitors, SGLT-2 inhibitors, antiplatelets, anticoagulants and antiarrhythmic therapy prescribed to the patients. The mainstays of antiarrhythmic therapy were Class I-C antiarrhythmics, namely flecainide and propafenone, and Class III antiarrhythmics, namely amiodarone. Additionally, each patient’s self-reported adherence to the pharmacologic therapy was encoded.

All data were anonymously collected in a predesignated electronic case report form (eCRF). This form was created through the online RedCap platform (Vanderbit University), facilitating simultaneous and secure data input. The trial was conducted in accordance with the 1975 Declaration of Helsinki, and the study protocol was approved by the local ethical committee of each participating institution. The anonymity and confidentiality of the collected data were ensured in accordance with the applicable legislation.

In this substudy, we included patients with known AF without mechanical prosthetic valves and moderate-to-severe mitral valve stenosis. In this report, we focus on the anticoagulation status of hospitalized AF patients. We included all the oral anticoagulation agents which were available in Greece (four vitamin K antagonists (acenocoumarol)). Patients with prior stroke, history of major bleeding, poor adherence to anticoagulation therapy and end-stage renal disease (eGFR < 15 mL/min/1.73 m^2^) were considered candidates for left atrial appendage closure.

Qualitative data are presented with absolute and relative frequencies (%) and compared with Chi-square tests of independence. Quantitative data are presented with means and standard deviation values or medians and first–third quartiles. The statistical significance level was set at 5% (a = 0.05), and a two-tailed analysis was carried out using the SPSS 20.0 software package (SPSS Inc., Chicago, IL, USA).

## 3. Results

The total sample of the HECMOS study consisted of 918 patients from 55 hospitals with a nationwide distribution. Two hundred-fifty-six patients (mean age 76.6 ± 11.7, 148 males) fulfilled the criteria of non-valvular AF and were included in our analysis. Most of them (*n* = 159; 62.1%) suffered from permanent AF (Table 1). The mean CHA2DS2-VASc score was 4.28 ± 1.7, while the mean HAS-BLED score was 1.47 ± 0.9. Among the study population, smoking status, defined as current tobacco use, was observed in 7.4% of patients, while 88 patients (34.4%) suffered from comorbid diabetes mellitus. A considerable proportion of the study subjects were hypertensive (73.8%), which is consistent with the high rates of chronic kidney disease (38%). Severe valvopathies, either aortic stenosis or regurgitation, were present in 4.7%, while severe mitral insufficiency occurred in 3.5%.

The primary cause of hospital admission was the deterioration of heart failure status and less commonly acute heart failure symptoms (in total, more than half of the patients (54.3%)). Most patients with AF and heart failure suffered from the persistent type of AF. AF paroxysms were the second most common cause for hospital admission among patients with known AF (14.1%), while bradyarrhythmic reasons were present in 7.3%. Acute coronary syndrome was the cause of admission for 6.5%. 

AF ablation was performed in nine (3.4%) of the study participants. Rhythm control through electrical cardioversion was attempted in 34 (13.2%) patients. Regarding cardiac implanted electronic devices, 27 patients had undergone permanent pacemaker implantation, 9 patients (3.5%) had a cardioverter–defibrillator implanted, and CRT-D was utilized in 1.5% of the participants. No CRT-P therapy was recorded in the dataset. Additionally, none of the study participants had undergone atrioventricular node ablation and subsequent pacing. 

We sought to analyze the anticoagulation status, stratified by the CHA2DS2-VASc score. Only three patients had a CHA2DS2-VASc of 0 or 1 (female), and they all inappropriately received anticoagulants [3]. Sixteen out of eighteen patients with a CHA2DS2-VASc score of 1 or 2 (female) were also treated with anticoagulants. Among 235 patients with a CHA2DS2-VASc score > 1 or >2 (for females), only 33 (14%) did not use anticoagulants (nine had declined it, and there had been no relevant recommendation in the other twenty-four). Surprisingly, there were no statistically significant differences regarding baseline characteristics with those appropriately anticoagulated—although the small size of this subgroup may have precluded the detection of any divergent pattern. Regarding the patients under anticoagulant therapy (*n* = 221), 191 (86.4%) received non-vitamin-K antagonist oral anticoagulants (NOACs), and 30 (13.6%) received vitamin K antagonists (acenocoumarol) (Figure 1).

Relative indications for left atrial appendage closure were present in 68 patients with NVAF (63 had only one risk factor, 5 had two and none had three or four concurrent risk factors. In detail, 36 had a prior stroke, 17 patients had a history of major bleeding, 15 patients reported poor or no adherence to the anticoagulant therapy and 5 had an eGFR value < 15 mL/min/1.73 m^2^. Moreover, 33 had a HAS-BLED score ≥ 3. No left atrial appendage closure procedure was reported (Figure 2).

## 4. Discussion

Anticoagulation status was nearly optimal in a high-thromboembolic-risk population of cardiology patients who were mainly treated with NOACs. Our study also identified that one out of four AF patients should be screened as a potentially eligible candidate for left atrial appendage closure.

Adherence to anticoagulant agents was always considered a critical issue since almost half of the AF patients with a high risk of stroke were undertreated in the coumadin era [14,15]. Overall, anticoagulation rates increased to 60% among eligible AF patients in the first years of the use of NOACs in North American registries and to 80% in European registries [16,17,18]. The MISOAC-AF was the first Greek trial that evaluated the real-life anticoagulation prescriptions of AF patients upon tertiary hospital discharge [5]. A total of 768 unselected patients with nonvalvular AF who were discharged between December 2015 and November 2017 had similar mean CHA2DS2-Vasc (4.4) and HAS-BLED (1.9) scores. Among the patients at significant stroke risk, 14.6% were not prescribed an OAC in absolute accordance with our results [5]. In this direction, almost all patients with a Class IIa indication for an OAC (a CHA2DS2-VASc score of 1 or 2 (female)) received anticoagulants. The suboptimal use of anticoagulation therapy in eligible patients is attributed to older age, bleeding risk, side effects and noncompliance [7].

Our sample was limited (*n* = 3) regarding low-risk stroke patients, and all of them were administered an OAC. Along these lines, 53% of AF patients with a CHADS score of 0 in the first year of receiving an NOACs received an OAC inappropriately [18]. Clinicians may be anticoagulating these patients simply because they have AF, regardless of their low rate of thrombotic events, although it is also possible that these patients may have been anticoagulated for conditions other than AF, such as valvular heart disease or venous thromboembolic events [19].

Anticoagulant agents increase bleeding risk [20] and, although the overall risk may be lower with NOACs when compared with warfarin, it is still nonzero [21]. Although left atrial appendage closure-related scientific societies’ recommendations are weak (IIb) [3,4], there is an increasing number of left atrial appendage closure procedures occurring worldwide [22]. Interestingly, there is no scientific consensus on the definitions of absolute or relative contraindications for OAC therapy for patients with AF and, consequently, on the exact indications for left atrial appendage closure. We selected potential indications for left atrial appendage closure based on the last expert consensus statement on catheter-based left atrial appendage closure [23]. First, patients with previous intracranial bleeding or stroke on adequate OAC treatment exhibited significant reductions in stroke/TIA and major bleeding events after left atrial appendage closure, without compromising safety [24,25]. Patients with severe renal dysfunction (eGFR < 15 mL/min) constitute a specific AF population since the routine use of NOACs should be avoided and warfarin is harmful if anticoagulation control is poor [26]. Even in patients with a GFR of less than 15 mL/min, left atrial appendage closure significantly reduced the stroke/TIA rate and major bleeding in comparison with the expected annual risk [27]. The nonadherence of patients and clinicians is also an important issue, whereas compliance with treatment is crucial, especially with NOACs since these drugs have relatively short half-lives [28].

Based on the above indications, we suggest that one out of four hospitalized patients with AF might be considered a potential candidate for left atrial appendage closure. If we take into account the relatively stronger indications for left atrial appendage closure (a prior major bleeding event or prior stroke), then one out of five patients in our study were eligible in accordance with a previous Greek study in which one out of six patients with AF had these strong indications for left atrial appendage closure [12]. Furthermore, in MISOAC-AF, 1 out of 10 patients was eligible for left atrial appendage closure according to looser criteria exclusively, such as a high HAS-BLED score (≥3) or the existence of end-stage renal disease (eGFR < 15 mL/min/1.73 m^2^) [12]. The proportion of potential candidates was one out of eight in the present study when considering these looser indications.

Nonetheless, it cannot be claimed that 25% of hospitalized AF patients should undergo an invasive procedure which is not devoid of complications [29]. This becomes more relevant given the imminent arrival of the next generation of anticoagulants that focus on factor XI, promising reduced bleeding (factor XI has a greater role in the etiopathogenesis of thrombosis than in physiological hemostasis) without compromising the anticoagulation effect [30]. Additionally, one might suggest that in the future, based on randomized trial results, the discontinuation of anticoagulation therapy after a successful ablation procedure will be feasible without any increase in the risk of stroke, altering current practice [31].

## 5. Limitations

The generalizability of our results is hampered by our study’s snapshot design. Our study included patients admitted to a cardiology ward for any reason with coexisting AF. Therefore, selection bias cannot be excluded as the trial population is not representative of a typical outpatient with AF who likely experiences fewer comorbidities; however, it reflects a typical clinical practice population with AF. The snapshot design offered the opportunity to gather abundant information regarding participants and index admission characteristics in a short period of time; however, there is a paucity of follow-up data. Follow-up data are crucial to estimating the influence of individual clinical variables on adverse outcomes, as well as the impact of therapeutic interventions. Similarly, the cross-sectional nature of the study does not reflect seasonal or other potential variations throughout the year, despite the representative epidemiological picture of Greece for the selected population.

## 6. Conclusions

Anticoagulation status appears to be nearly optimal in a high-thromboembolic-risk population of cardiology patients who are mainly treated with NOACs, reflecting increases in clinician commitment and patient adherence. We also identified that one out of four AF patients should be screened as potentially eligible candidates for left atrial appendage closure, always taking into consideration the absence of large randomized clinical trials comparing left atrial appendage closure with NOAC therapy. Until then, a multidisciplinary team approach is required to choose patients who may benefit more from this invasive therapy.

## Figures and Tables

**Figure 1 medicina-59-01881-f001:**
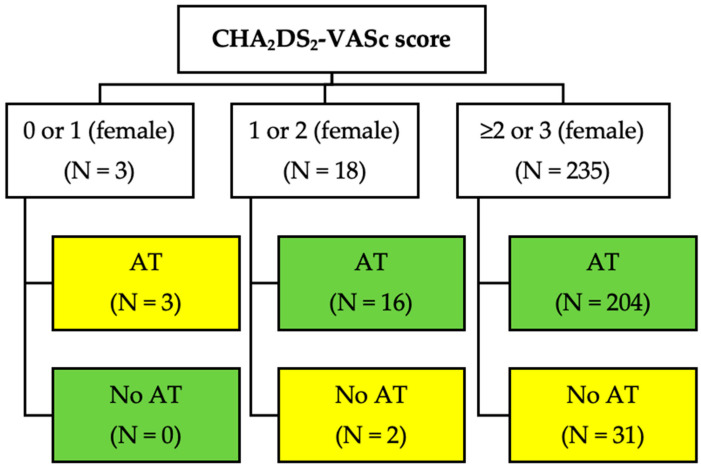
Anticoagulation therapy in different thromboembolic-risk AF patients according to the CHA2DS2-VASc score (AT: anticoagulation therapy).

**Figure 2 medicina-59-01881-f002:**
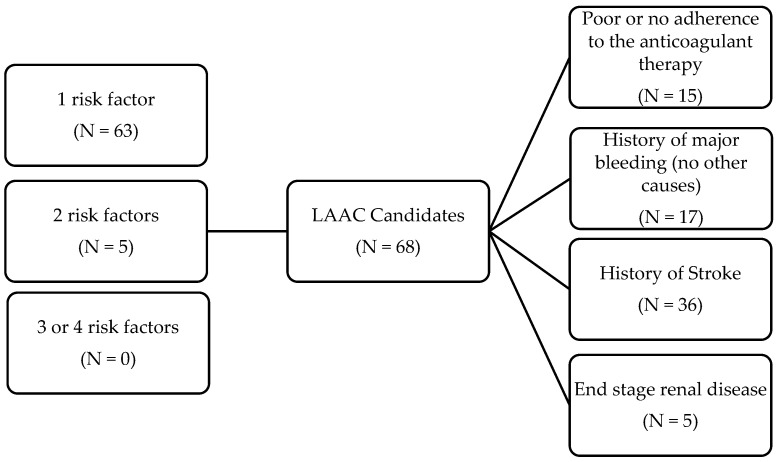
Potential candidates for left atrial appendage closure (LAAC). Causes for LAAC indication (third column) are non-mutually exclusive, i.e., five patients had two factors concomitantly (three experienced major bleeding plus end-stage renal disease and two experienced major bleeding plus stroke while on anticoagulant medication).

**Table 1 medicina-59-01881-t001:** The study population’s baseline demographics and clinical characteristics. Poor INR control was defined as the time in therapeutic range (TTR) < 70% [3].

History of Atrial Fibrillation	
Classification (*N* = 256)	
Paroxysmal	97 (37.9%)
Permanent	159 (62.1%)
On antiarrhythmic medication (including β-blockers)	200 (78.1%)
Previous cardioversion	34 (13.2%)
History of catheter ablation	9 (3.6%)
Permanent pacemaker	27 (10.5%)
Defibrillator	9 (3.5%)
CRT-D	4 (1.5%)
Comorbidities (*N* = 256)	
Smoker (current)	19 (7.4%)
Hypertension	189 (73.8%)
Diabetes	88 (34.4%)
Chronic kidney disease	97 (38%)
Obstructive sleep apnea	13 (5.1%)
Severe aortic stenosis	10 (4%)
Severe aortic regurgitation	2 (0.7%)
Severe mitral regurgitation	9 (3.5%)
Anticoagulation treatment among patients with good compliance (*N* = 223)	
VKA—good INR control	25 (11.2%)
VKA—poor INR control	7 (3.1%)
Dabigatran	27 (12.1%)
Rivaroxaban	53 (23.8%)
Apixaban	111 (49.8%)
Beyond anticoagulants, are on any other medical treatments which increase bleeding risk? (Yes) (*N* = 256)	43 (16.8%)

## Data Availability

The data that support the findings of this study are available from the corresponding author upon reasonable request.

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
