# Peer review of "Anticoagulation Status and Left Atrial Appendage Occlusion Indications in Hospitalized Cardiology Patients with Atrial Fibrillation: A Hellenic Cardiorenal Morbidity Snapshot (HECMOS) Sub-Study"

_medicina, 2023, doi:10.3390/medicina59101881_

Round 1
Reviewer 1 Report
1. Mention how you define good and poor VKA control in Table 1.
2. Emphasize the limitations of this study.
3. Include in Table 1 the patient demographics, comorbidities, clinical characteristics, complications of anticoagulant therapy, etc.
4. Is it possible to report adherence to the anticoagulant therapy?
5. Rephrase lines 180 to 185; it is difficult to understand.
1. Rephrase lines 180 to 185; it is difficult to understand.
Author Response
- Mention how you define good and poor VKA control in Table 1.
We have now included the definition of poor INR control, based on the 2020 ESC Guidelines on Af.
- Emphasize the limitations of this study.
A separate dedicated section has been created, with improvements in expression.
- Include in Table 1 the patient demographics, comorbidities, clinical characteristics, complications of anticoagulant therapy, etc.
We have significantly expanded Table 1, enriching it with the requested data. Regarding anticoagulation complications, the 17 patients who had suffered major bleeding had no other obvious causes than anticoagulation.
- Is it possible to report adherence to the anticoagulant therapy?
There are relevant numbers in the Table, however more detailed information (e.g., number or frequency of missed doses) is unavailable.
- Rephrase lines 180 to 185; it is difficult to understand.
This paragraph has been restructured to improve its comprehensibility.
Reviewer 2 Report
The authors present a descriptive snapshot of anticoagulation status and LAAC indications in a population of patients with AF hospitalized in cardiology departments. The topic is interesting, but the paper requires some clarifications.
1. To assess the generalizability of the findings, study population should be better defined. Table 1 should include other data such as reason for admission, main comorbidities (hypertension, coronary artery disease e.g.) and current antithrombotic or antiarrhythmic therapies.
2. It would be interesting to have a better description of the population with CHA2DS2-VASc > 1 who is not anticoagulated. What is the reason why they have not been anticoagulated? Was it a decision of the patient or of the physician? Do they have different baseline characteristics compared to anticoagulated patients?
3. Authors should clarify why they consider the 3 patients with CHA2DS2-VASc 0 or 1 to be inappropriately anticoagulated. As properly discussed later in the text, even patients why low CHA2DS2-VASc score may have adequate indications to anticoagulation (e.g. VTE or, simply, AF lasting > 48 hours with a planned cardioversion).
4. It is not clear how indications to LAAC have been calculated: it is stated that 60 patients had 1 risk factor and 4 had 2 risk factors, which should sum up to a total of 68 single risk factors. On the contrary, the sum of the single risk factors (“36 had a prior stroke, 17 patients had a history of major bleeding, 15 patients reported 124 poor or no adherence to the anticoagulant therapy and 5 had eGFR< 15 ml/min/1.73m2”) adds up to 73. Additionally, there is inconsistency in the number of patients with 2 concurrent indications to LAAC: 2 in line 34 and 4 in line 123.
5. Phrasing in lines 132-133 (and line 204) is not clear. Ideally, all patients with AF should be screened for LAAC eligibility. To my understanding, the presented data, rather than identifying “patients that should be screened as potentially eligible candidates for LAAC.”, identifies those that indeed are potentially eligible candidates.
6. Lines 184-184 require a citation.
Many corrections needed. Sentence in lines 180-183 is very hard to understand.
Author Response
- To assess the generalizability of the findings, study population should be better defined. Table 1 should include other data such as reason for admission, main comorbidities (hypertension, coronary artery disease e.g.) and current antithrombotic or antiarrhythmic therapies.
We have significantly expanded Table 1, enriching it with the requested data.
- It would be interesting to have a better description of the population with CHA2DS2-VASc > 1 who is not anticoagulated. What is the reason why they have not been anticoagulated? Was it a decision of the patient or of the physician? Do they have different baseline characteristics compared to anticoagulated patients?
Please see lines 114-118 were the requested information has been added to the manuscript.
- Authors should clarify why they consider the 3 patients with CHA2DS2-VASc 0 or 1 to be inappropriately anticoagulated. As properly discussed later in the text, even patients why low CHA2DS2-VASc score may have adequate indications to anticoagulation (e.g., VTE or, simply, AF lasting > 48 hours with a planned cardioversion).
Based on the 2021 ESC Guidelines on management of AF (figure 12, page 408), those at “low stoke risk – CHA2DS2-VASc score 0m/1f) do not need anticoagulation. Although the wording “inappropriately” is stronger, we consider this appropriate because the cost/benefit ratio is definitely unfavorable to the anticoagulation approach in this group.
- It is not clear how indications to LAAC have been calculated: it is stated that 60 patients had 1 risk factor and 4 had 2 risk factors, which should sum up to a total of 68 single risk factors. On the contrary, the sum of the single risk factors (“36 had a prior stroke, 17 patients had a history of major bleeding, 15 patients reported 124 poor or no adherence to the anticoagulant therapy and 5 had eGFR< 15 ml/min/1.73m2”) adds up to 73. Additionally, there is inconsistency in the number of patients with 2 concurrent indications to LAAC: 2 in line 34 and 4 in line 123.
We apologize for the conflicting numbers. We have revamped Table 1 as well as Figure 2 to include the actual data figures. More specifically, the first column refers to patients as individuals (i.e., 63 with 1 RF + 5 with 2RFs = 68 patients), whilst the third column reports the number of RFs, which are not mutually exclusive, thus their total is 68 + 5 (as in 5 patients with a second concomitant RF) = 73.
- Phrasing in lines 132-133 (and line 204) is not clear. Ideally, all patients with AF should be screened for LAAC eligibility. To my understanding, the presented data, rather than identifying “patients that should be screened as potentially eligible candidates for LAAC.”, identifies those that indeed are potentially eligible candidates.
Indeed, we have rephrased this as: “are potentially eligible…”
- Lines 184-184 require a citation.
Indeed, we have added the citation of the ongoing OCEAN trial design, bound for completion in mid-2024.
Round 2
Reviewer 1 Report
None
Reviewer 2 Report
I thank the authors for having properly addressed all my questions.
Quality of english has been improved